# Combined Assessment of Preschool Childrens’ Exposure to Substances in Household Products

**DOI:** 10.3390/ijerph16050733

**Published:** 2019-02-28

**Authors:** Joo-hyon Kim, Kwangseol Seok

**Affiliations:** Division of Chemical Research, National Institute of Environmental Research, Hwangyeong-ro 42, Seo-gu, Incheon 22689, Korea; aped@korea.kr

**Keywords:** household product, preschool children, combined exposure, exposure assessment, product ingredients

## Abstract

Understanding how indoor-air contaminants affect human health is of critical importance in our developed society. We assessed the combined exposure by inhalation of preschool children and children to household products. A total of 1175 families with 72 infants, 158 toddlers, 230 children, and 239 youths were surveyed to determine the combined respiratory exposure concentrations and amounts associated with 21 substances in eight household product groups. We determined the mean concentrations of these substances in each product, and derived reference toxicity values based on the information gathered in order to identify respiratory health risks. On average, cleaners were used at a rate of 1.0 × 10^3^ g/month, while coating agents and other substances were used at 43 g/month. The combined inhalation exposure concentrations of methanol to infants and toddlers were 5.1 and 4.2 mg/m^3^ per month, respectively, with values of 2.1 and 1.7 mg/m^3^ for isopropanol, respectively. Risks to preschool children and children should be assessed on the basis of the toxicity values of combined exposed hazardous substances, as well as their combined concentrations and amounts. This exposure assessment approach can be used to establish improved guidelines for products that may pose inhalation hazards to preschool children and children.

## 1. Introduction

Indoor air contains numerous substances that are emitted from multiple household products [1,2]. Consequently, consumers of household products are exposed to several types of substance on a daily basis. Indoor air has been postulated to play an important adverse role, contributing to persistent wheezing because many people, especially mothers with small children, spend most of their days indoors [3]. A variety of substances that are widely used as solvents, preservatives, odor agents, biocidal active ingredients, and additives in common household products are potentially hazardous [4]. These substances can affect human health because they are potentially carcinogenic [5], allergenic [6,7,8], and impact adversely through inhalation [2,9]. The development of respiratory hypersensitivity or asthmatic symptoms in some susceptible individuals has been associated with the indiscriminate use of household products and exposure to specific substances [10].

The chemicals used in household products were in a “grey area” in Korea from a chemical management perspective because many of these chemicals were registered as “existing chemicals” and were used without any chemical risk assessment [11]. For example, disinfectants were used in humidifier water tanks in order to prevent the growth and spread of germs, molds, and/or algae. Recently, the inhalation of aerosolized water from a humidifier that contained disinfectants led to serious lung issues that resulted in 200 deaths and 700 injuries, including children [12]; this tragedy heightened public attention and raised concerns. In 2013, the Korean government enacted regulations on substances in a variety of household products that may adversely affect health in an effort to promote the safe use of these products (The Korean Registration, Evaluation, Authorization and Restriction of Chemicals or K-REACH)) [11]. Household products that fall under these regulations are required to be assessed for exposure and risk in order to evaluate the health and environmental hazards associated with their use. Exposure and risk assessment involves the systematic scientific characterization of potential adverse health effects resulting from human exposure to hazardous chemicals or situations [12,13,14,15,16,17,18].

Recently, several studies on the exposure of infants and toddlers to phthalates and other substances in household products have been published [19,20]. Particular cases include the recent trend of incorporating antimicrobial ingredients in a multitude of otherwise traditional household products, and apparent increases in the environmental impact of many active ingredients used in personal care and household products, and pharmaceuticals [4]. The exposure of children to environmental contaminants is expected to be different to, and in many cases much higher than, that of adults. In general, humans in their developmental stages (fetus, infant, toddler, and child) appear to be more severely affected by exposure to contaminants than adults [21]. While exposure and risk assessments of household products for infants, toddlers, and children are being extensively pursued in Korea, they remain insufficient (e.g., exposure patterns of diverse stressors, use patterns of products, reliable exposure factors for risk assessment study, and combined and (or) cumulative exposure information by preschool children and children). In previous studies, the irritant and respiratory-health effects of deodorants and air-fresheners were assessed in relation to the dermal and inhalation exposure of adult consumers [22,23].

Herein, we examined the concentrations of substances in household products and assessed the combined use of these products, as well as combined inhalation exposure to infants and toddlers based on “the worst-case scenario”. This study also presents an approach for compiling common assessment principles for combined exposure to household products. Exposure assessments were carried out in order to characterize real-life situations involving infants, toddlers, children, and youths, in which: (1) potentially exposed populations were identified, (2) potential exposure pathways identified, and (3) the magnitude, frequency, duration, and temporal contact patterns involving a particular chemical (potential dose) quantified [24]. This comprehensive study provides motivation for the mandatory regulation of household products.

## 2. Materials and Methods

### 2.1. Survey

A market searching was conducted to elucidate which studied household products were commonly used in the Korean market. Survey-company carried out the market searching of the currently used household products. The web searching collected current information on the products. In our previous study, in order to obtain data on the exposure of consumer to household products, web surveys were carried out [25]. The purpose of the web-surveys was to obtain information about the household exposure of the studied products at home. The survey was carried out a Korean survey company. The survey-company had a participant pool in all provinces and cities in Korea. At first, an e-mail was sent to 12 times ~ 15 times the number of targeted participants (1000). If participants agreed to take the survey and answered that they had experience using at least one product among the studied products, a web link was sent to them. The web-survey questionnaires consisted of purchasing/using information of products as follows: list of products used at home and the frequency of use, estimation of quantitative duration of exposure to products, quantitative amount of products used, and demographic data. Furthermore, the survey questionnaire included questions on family members of the respondent and on combined multiple-use of products. Combined exposed amounts of the studied families by household products at home were evaluated. In total, 1175 families completed the questionnaire. All respondents were actual consumers of the studied product groups. To ensure that the study population was representative of the Korean population, gender-balanced participants with equal age-group distributions were selected [26,27,28].

On the basis of the survey results, we determined the combined exposure to infants and toddlers of hazardous substances from household products. In this study, eight product groups were selected on the basis of their frequency of use; the target household products have been established by the Korean Ministry of Environment (KMOE) as products of concern based on risks to residential consumers (Table 1) [29]. According to the results of our previous study [25], the web survey questionnaires were used to determine combined exposure to household products by the studied families, and were divided as follows: (1) information on the retention and use of products, namely lists of retained target products, amounts of retained products, product use/purchase periods, and qualitative descriptions of product co-use habits; and (2) information on demographics, namely ages of family members, average time spent indoors for each member (separately for working and non-working days), occupation (employed, unemployed, student, housewife, or other), type of house, and average house size. Single-product-use exposure data were obtained from national survey data (Korean consumer exposure factors to household products) [26]. Furthermore, seven age groups were constructed for the survey, namely, families with infants (0–2 years), toddlers (3–6 years), young children (7–9 years), children (10–12 years), young youths (13–15 years), youths (16–18 years), and adults (19 years and older); these groups are based on the long-term resting inhalation rates (m^3^/day-kg, weight adjusted) for children from birth to 18 years of age [30,31]. To determine the combined exposure times for inhalation uptake, age-dependent indoor times per family were surveyed against short-duration (below 8 h), medium-duration (8–16 h), and long-duration (16–24 h) times; the average indoor times per day, week, and month were divided by age and occupation group. The distribution of indoor time per day was also summarized under working and non-working days. The distribution per week included five working and two non-working days, while the distribution per month was calculated as 22 working and eight non-working days.

### 2.2. Calculating the Combined Exposure Concentration

Exposure factor values for each household product (single consumer use) were used to calculate the combined exposure concentration and the amount of multiple use of a household product, based on usage and application type. Reliable exposure factors, including frequency of use, amount of use per application, and duration of use by end-users, were established according to National Institute of Environment Research data (Table A1) [32]. Korean exposure factors are specified by law, and exposure scenarios for household products based on inhalation routes are also specified, with different countries exhibiting different exposure factors. On the basis of these exposure factors, the amount of exposure to a target product group per month was calculated. The application types of each product group were selected based on the ubiquitous indoor respiratory exposure of the studied subjects; these types were: trigger-foam and liquid for toilet and bathroom cleaners, trigger for glass cleaners, and others (Table 2). 

Considering the worst-case scenario, the combined exposure by adults in families that use multiple product groups was assumed to be the same as the combined exposure amounts by infants, toddlers, children, and youths. However, in order to avoid overestimating these amounts, the combined amounts by application per month were based on the summations of mean product consumptions instead of high-percentile amounts. Exposure factors were selected by considering the product application type, frequency of use, duration of use, and amount of use per application. In this study, seven application types of cleaners, two application types of bleaching agents, two application types of coating agents, two application types of adhesives, three application types of air fresheners, and five application types of deodorants were selected (Table 2).

### 2.3. Reference Toxicity Values for Substances in Products

Official toxicology reports and studies (i.e., U.S. EPA documents, U.S. California EPA documents, NIH reports, the U.S. RED report, and the E.U. ECHA Dossier) were used for each substance in order investigate its toxicity value in the various products used [33]. In order to investigate the inhalation toxicities of the various substances, inhalation reference concentrations (RfCs) based on information used by the California EPA [34] and U.S. EPA [35,36] were determined for eight substances. In particular, we derived reference toxicity values based on chronic NOAELs (no observed adverse effect levels) and uncertainty factors from NIH reports, the U.S. RED report, the EU ECHA Dossier, and others. Reference toxicity values (i.e., chronic NOAECs and NOAELs) were calculated according to official guidelines; uncertainties in the extrapolation of experimental data to real human exposure situations, inter-species and intra-species differences, differences in exposure durations, and differences in toxicological values (i.e., LOAELs and NOAELs) were taken into consideration [30]. Toxicity values, assessment factors, reference values, and the target margins of exposure (MOEs) are presented in Table 3.

### 2.4. Analyzing Substances in Products

In this study, 21 substances were selected based on information on the proportions of ingredients used in the various product groups provided by Korean manufacturing companies. The substance-analysis method depended on the characteristics of each product group; there were 14 substances in the cleaning category, seven in bleaching agents, 11 in coating agents, 11 in adhesives, 10 in air fresheners, and 10 in deodorants, with different application types (Table 4 and Table 5).

In all substance analyses, the standard operating procedure developed by the National Institute of Environment Research (NIER) was followed [29], and adherence to quality-assurance/quality- control requirements was maintained, including method and reagent blanks, instrument detection limits, and calibration curves. The limits of quantitation (LOQs) of the substances are presented in Table 4 and Table 5. LOQ was defined as the lowest concentration at which the analyte can not only be reliably detected at which some predefined goals for bias and imprecision are met; concentrations below the LOQ were assigned a value of zero during data analyses.

### 2.5. Worst-Case Assessment of Combined Exposure

Concentrations of substances in the air, and combined exposure concentrations and amounts through inhalation were calculated using exposure equations provided in the NIER notification [30,32]. In order to avoid overestimating the detection ratios of substances in product groups, the mean values were used as the concentrations of the substances in the various products. During inhalation-exposure assessment, the combined exposure concentration through inhalation was calculated as follows:

Concentration of air (C_a_), mg/m^3^
C_a_ = (A_p_ · W_f_/V · N) · [1 − exp(−N · t)]/t(1)

Inhalation concentration (C_inh con_), mg/m^3^
C_inh con_ = C_a_ · fabs · t/(24 h·30 d) · AF(2)

Inhalation amount (C_inh amount_), mg/m^3^
C_inh amount_ = C_a_ · IR · fabs · t/BW(3)
where A_p_ is the summed amount of use per product in each product group; i.e., cleaning agents (C-1, C-2, C-3, C-5, C-6, C-8, and C-12), bleaching agents (B-1 and B-3), coating agents (CA-1 and CA-2), adhesives (AD-1 and AD-2), air fresheners (AF-1, AF-2, and AF-3), and deodorants (D-1, D-2, D-3, D-4, and D-5) (Table 2, Table 6). t is the time for which users are exposed to products. 

W_f_ is the proportion of the mean concentration of a substance in a product, V is the assumed total volume of a home (97.4 m^3^), which is the mean size of a home in Korea as reported by the Korean consumer exposure factors handbook by considering places where family members use the target product groups, while N is the ventilation rate (h^−1^), which Bremmer and Veen give for a non-specified room [31], is taken as a default value (0.6 h^−1^) and calculated to be the total home to general ventilation rate per month. We estimated that the time for which users are exposed to products is the exposure duration in the place where the product was used. t is calculated to the mean time that the subjects stayed inside their homes on working and non-working days per month, on the basis of the survey results (h/month); i.e., 16 h per working and non-working days for infants, 12 h per working day and 16 h per non-working day for toddlers, and 8 h per working day and 16 h per non-working day for children and youths. Fabs, which is the fraction absorbed by the body, is assumed to be 1 (100%). The AF (air-borne fraction, 1) is assumed to be unity according to the NIER notification. BW (body weight, kg) and IR (resting inhalation ratio, m^3^/day-kg) values per age were sourced from the Korean Exposure Factors Handbook and Korean Exposure Factors Handbook for Children [39,40]. In addition, we compared the calculated inhaled combined exposure data between infants, toddlers, children, and youths (Table 6).

## 3. Results

### 3.1. Distributions of Products, Infants and Toddlers in Families

We obtained questionnaire data from 1175 families. The survey data showed that 65 families had 72 infants, 136 families had 158 toddlers, 109 families had 114 young children, 105 families had 116 children, 98 families had 105 young youths, and 123 families had 134 youths (Table 7). The surveyed families retained and regularly used nearly all of the studied product groups. Families with infants, toddlers, children, and youths also used high proportions of the studied product groups (Table 7). Table 1 reveals that families with infants, toddlers, children, and youths used the studied products, and potentially combined exposed infants to these products through regular use. Families with infants, toddlers, children, and youths did not show the prevalence for the use of the studied products. In order to calculate age-dependent combined exposure, each studied age category was divided according to the long-term resting inhalation rates of Korean subjects, such as those of infants (male: 0.87 m^3^/day-kg, female: 0.9 m^3^/day-kg), toddlers (male: 0.57 m^3^/day-kg, female: 0.53 m^3^/day-kg), and others (Table 7). To estimate combined exposure times, we separately surveyed age-dependent indoor times in homes on working and non-working days.

The numbers of working and non-working days were calculated to be 21 and 9 d per month, respectively. In case of adults, the indoor times on working and non-working days differed between employed and unemployed individuals, as well as males and females. However, the indoor times of infants and toddlers were not significantly different between the surveyed families.

### 3.2. Combined Amounts of Products Used

To calculate the combined amounts of products used per month, we summed the amounts of all products used per month in the various product groups after calculating the use of each single product. On the basis of the survey results, target products with potential for inhalation exposure, and their application types, were selected. Regarding the cleaners, the main inhalable applications were the trigger-foam type and liquid type for toilet and bathroom use, trigger type for glass, trigger-foam type and liquid type for kitchens, trigger-foam type for mold and moss, and aerosol-spray type for air-conditioners (Table 1). The products and the application types of other selected product groups are listed in Table 1. The total amount of each selected product group used per month was calculated according to the exposure factors listed in Table 2; the combined amounts used per month differed among the various products. The mass per use was highest for the cleaner group and lowest for deodorants (aerosol fabric sprays). 

The mean amount of products used in the cleaner group was determined to be 1.0 × 10^3^ g/month (3.5 × 10^2^ g/month for the bleaching agents, 4.3 × 10 g/month for coating agents, 2.8 g/month for adhesives, 2.9 × 10^2^ g/month for air fresheners, and 3.4 × 10^2^ g/month for the deodorant groups) (Table 2).

### 3.3. Toxicity Information the Studied Substances

RfCs and chronic NOAELs were determined through a brief overview of previous toxicity studies aimed at estimating risks (Table 3); toxicity values are based on long-term substance exposure and inhalation-exposure routes. RfC values determined by the U.S. EPA and Cal/EPA were used to evaluate the toxicity characteristics of toluene, methanol, 2-butoxyethanol, methyl isobutyl ketone, ethylbenzene, xylene, isopropanol, and benzene. ECHA registration dossiers and U.S. EPA RED reports were used to determine the toxicity values of benzyl alcohol, ethyl acetate, 1-butanol, and others. These reference toxicity values were used to identify the risks to human health of the studied substances. To assess the health risks to infants and toddlers at the levels of exposure, the margin of exposure (MOE, a reference point derived from the dose-response relationship) and the hazard quotient (HQ, the ratio of the substance exposure level to a reference toxicity value) need to be calculated for each substance [33]. If the MOE of a substance is lower than the target margin of exposure (MOE), and the HQ ratio is greater than unity, the exposure level of the substance may pose a health risk [41]. In this study, the detection ratios of the substances in the various products were relatively low. As a further study, we are investigating health risk assessment study to preschool children and children from household product use by primary users.

### 3.4. Analyzing Substances in Product Groups

The measured concentrations of substances (mg/kg) in the product groups of all application types are listed in Table 4 and Table 5. Among the 14 substances, 13 were detected in 109 or 1421 cleaning products at detected numbers of 1–24 products; propylene glycol was detected at the highest levels (in 24 out of 109 products). Toluene, methanol, 2-methyl-4-isothiazolin-3-one, 2-*n*-octyl-4-isothiazolin-3-one, and didecyldimethylammonium chloride were detected in less than five of the 109 cleaners. Of the 13 substances in the cleaners, five exhibited maximum concentrations greater than 10,000 mg/kg. In case of the bleaching agents, seven substances were found or not detected in 13 or 100 bleaching agents, respectively. 11 substances were found in 43 coating agents and 11 more substances were found in 30 adhesive products. Maximum concentrations of ethyl benzene, xylene, and isopropanol in coating agents exceeded 10,000 mg/kg. Among the 11 substances, nine were detected in adhesive agents at low rates. The maximum concentrations of toluene and methyl isobutyl ketone were relatively high in the adhesive agents. Only four substances were rarely detected in air fresheners; the maximum concentrations of isopropanol and 2-phenoxyethanol in deodorant products exceeded 10,000 mg/kg. The levels of these substances and the analyzed maximum and minimum concentration ranges are summarized in Table 4 and Table 5.

### 3.5. Combined Exposure Amounts by Inhalation

On the basis of the measured mean substance concentrations in five product groups, we calculated the combined exposure concentrations and amounts of substances in the various products inhaled by infants, toddlers, children, and youths per month. A summary of the worst-case combined exposure assessments for infants, toddlers, children, and youths by inhalation is summarized in Table 7. The studied products and application types were C (C-1/2/3/5/6/8/12), CA (CA-1/2), AD (AD-1/2), AF (AF-1/2/3), and D (D-1/2/3/4/5). The calculated combined concentrations and amounts of substances that infants and toddlers were exposed to are higher than those calculated for children and youths. The highest combined exposure concentrations and amounts of substances inhaled by infants and toddlers are 5.1 mg/m^3^/month and 3.1 × 10 mg/kg/month (methanol for infants), 2.1 mg/m^3^/month and 1.2 × 10 mg/kg/month (isopropanol for infants), 4.2 mg/m^3^/month and 7.3 mg/kg/month (methanol for toddlers), and 1.7 mg/m^3^/month and 3.0 mg/kg/month (isopropanol for toddlers). According to the ratios of the mean concentrations of substances in the studied products, the concentrations of these substances in the air were proportionally high, and combined exposure concentrations and amounts of substances inhaled were dependently high. Consequently, infants, toddlers, children, and youths are exposed to methanol, 2-butoxyethanol, methyl isobutyl ketone, ethyl benzene, xylene, isopropanol, propylene glycol, 2-phenoxyethanol, hexane, and heptane in high combined exposure concentrations and amounts.

## 4. Discussion

Inhalation is the predominant exposure route and is usually the result of airborne concentrations of substances in the breathing zones of exposed individuals, such as infants and toddlers. It may refer to active substances or to products in use and is expressed in mg/m^3^ as a time-weighted average concentration over a stipulated period of time. By its nature, this concentration provides an assessment of potential exposure [42]. Infants and toddlers are unintentionally more exposed to these substances by inhalation, due to relatively longer periods of time spent indoors in homes, than adults. Our calculations of the combined exposures of infants and toddlers to substances may be overestimates of the actual concentrations and amounts of exposure, considering the detection rates of the substances in the products studied. Therefore, in this study, we did not carry out a human-health risk assessment of the studied substances using the reference toxicity values in Table 4. As a further study, we are investigating health risk assessment study to preschool children and children from household product use by primary users. 

Exposure assessment involves determining concentrations to which humans may be exposed. The procedure for the human-health risk assessment of a substance involves comparing the exposure levels to which populations are exposed, or are likely to be exposed, to the exposure levels at which no toxic effects are expected to be observed. It should be noted that sub-populations might exist in any particular human population, and these may need to be considered individually when determining risk. Hence, exposure levels are derived separately for each relevant population/sub-population, and, where appropriate, different AELs or AECs (acceptable exposure level or concentration derived on the basis of threshold levels) are identified for critical endpoints, leading to the establishment of respective exposure-level/AEL or AEC ratios) [42]. 

Combined exposures are important components of current exposure- and risk-assessment protocols. When assessing exposure and health risk to infants and toddlers, exposure information from all potential exposure routes, including the air that infants breathe, should be cumulated. This study of combined exposure to substances in household products suggests that there are potential health-risk concerns for the studied subjects. Compared to adults, infants and toddlers stay at home for much longer periods of time and have high inhalation rates; they are therefore unintentionally more exposed to substances in multiple products used by adults. Children younger than 2 y spend the most time indoors at home, whereas older children spend the least amount of time indoors at home. Variability within each age category was substantial, but fair consistency was observed across all age categories (SDs of ~4 h). The indoor times of newborn infants was approximately 19.6 h, while that of three-year-old toddlers was ~18.0 h [24]. The combined inhalation exposure assessment was carried out progressively and the household-product groups managed by KMOE were selected for study. KMOE and NIER have conducted human risk assessments in order to assess hazardous ingredients, most of which are used in household products. Firstly, in order to calculate the combined exposure concentration and the amounts for infants and toddlers, we need to understand the current frequency of using products, and the mode of use in families with preschool children and children. To determine the combined exposure times of the products used, the average indoor times of infants and toddlers in homes need to be determined. The next step involves identifying target substances from among the ingredients contained in various products. Considering the components and mixing ratios provided by manufacturers in Korea, and the vapor pressures of the substances, we selected volatile ingredients and preservatives frequently used in household products. The RfCs or chronic NOAELs of the selected substances are calculated using available inhalation toxicity information. Estimating the potential risks associated with products is a necessary toxicological endpoint for the properties of the substances used. Selected substances in products were analyzed and mean concentrations in products were determined as inhalable concentrations to subjects in homes. Reliable exposure factors, including frequency of use, amount of use per application, and duration of use were established in a notification by NIER [43] (Table A1). As a last step in the assessment of combined exposure, product-induced concentrations of the studied substances in indoor air, and the combined exposure concentrations and amounts for the target subjects per month were calculated. 

This study assessed the respiratory effects of substances in products on preschool children and children. The “most representative worst-case scenario” reflects the worst cases for the evaluated products. The exposure- and risk-assessment approach provides improved guidelines for specific raw materials in household products that are known to have adverse health effects; equally importantly, it contributes toward setting limits for newly developed raw materials for household products that may pose threats to human health. In particular, although infants and toddlers do not purchase and use products, they are unintentionally exposed to several substances through a number of routes. In this study, the indoor concentrations of the substances were assumed to be calculated from equation 1 using the mean concentration of the substances in the various products. However, the actual concentrations of the substances in indoor should be analyzed according to the emitted amounts of these substances from the various products [2]. An understanding of the actual amount of studied substances in indoor air is needed for further studies to evaluate more refined combined exposure estimation.

## 5. Conclusions

This study investigated a fundamental approach to assess human exposure to household products used in daily life. The process of assessing exposure to household products used in home requires determining the patterns of use, exposure routes, and quantifying potential ingredients intake. This study determined the recent exposure factors of household products used by respondents group with preschool children, respondents group with children, respondents group with youths, and respondents group with only adults. To protect preschool children and children from several hazardous substances, more comprehensive exposure estimation and assumption are needed. More data are also required in order to reduce uncertainty in the assumptions and more accurate exposure estimation to ensure that substances are regulated appropriately. To improve the database available for assessing the exposure of infants, toddlers, and children, further research are required. The NIER in Korea provided guidance on exposure and risk assessment of substances in household products for consumers. This study followed the procedure of this guidance and focused on substances in household products consumed by households. The estimated amounts of combined exposure have the potential to influence infants, toddlers, and children. In the present study, respiratory effects of substances in household products on infants, toddlers, and children were aggregately assessed. The exposure scenarios used reflected the worst cases for the evaluated household products. Advances in emission analysis for substances will certainly improve the quality of exposure estimation for the assessment of health risk. Further studies are required in order to gain an understanding of the actual amounts of studied substances in indoor air so that more-refined combined exposure data are provided. More data are also required in order to reduce uncertainty in the assumptions made and to provide more-accurate exposure estimations that ensure substances are regulated appropriately.

## Figures and Tables

**Table 1 ijerph-16-00733-t001:** Distribution of products used by surveyed families.

Used Application Types of Product	Families with Infants (n ^a^)	Families with Toddlers (n ^a^)	Families with Children (n ^a^)	Families with Youths (n ^a^)	Adults only (n ^a^)	Description of Studied Products (Product Purpose and Application Type)	
C-1−C-12	58/65 (1)	116/136 (2)	182/214 (5)	198/221 (2)	549/721 (8)	Cleaner	Toilet & bathroom - trigger foam: C-1, liquid: C-2 Home glass - trigger: C-3 Vehicle glass - trigger: C-4 Kitchen - trigger foam: C-5, liquid: C-6, powder: C-7 Cleaning/removing mold & moss - trigger foam: C-8, gel: C-9 Drainpipe - liquid: C-10 Washing machine - powder: C-11 Air-conditioner leaning - aerosol spray: C-12
S-1−S-6	60/65 (4)	126/136 (6)	200/214 (9)	212/221 (5)	672/721 (14)		
B-1−B-3	15/65 (4)	33/136 (6)	51/214 (8)	68/221 (4)	160/721 (9)		
F-1−F-3	48/65 (3)	107/136 (3)	167/214 (8)	163/221 (2)	530/721 (10)		
CA-1−CA-6	27/65 (2)	35/136 (4)	58/214 (11)	66/221 (4)	152/721 (12)		
AD-1−AD-2	36/65 (4)	71/136 (11)	109/214 (18)	105/221 (12)	324/721 (34)		
AF-1−AF-3	38/65 (3)	67/136 (10)	114/214 (16)	119/221 (9)	401/721 (31)		
D-1−D-5	48/65 (3)	84/136 (5)	137/214 (8)	152/221 (5)	491/721 (13)		
C,S,B,F,CA,AD,AF,D	64/65 (1)	133/136 (2)	209/214 (5)	221/221 (0)	713/721 (7)	Synthetic detergent	Fabric and other use-powder: S-1, liquid: S-2, condensed liquid: S-3, tissue type: S-4 Woolen fabrics - liquid: S-5 Home dry cleaning-liquid: S-6
C,B,CA,AD,AF,D	64/65 (1)	130/136 (2)	204/214 (5)	219/221 (0)	675/721 (7)		
C,CA,AD,AF,D	64/65 (1)	130/136 (2)	204/214 (5)	219/221 (0)	675/721 (7)		
C,AD,AF,D	63/65 (1)	129/136 (2)	204/214 (5)	217/221 (0)	672/721 (7)		
C-1 or C-2	40/65	74/136	139/214	133/221	366/721	Bleaching agent	Fabric and other use – liquid: B-1, powder: B-2 Scrubbing by hand – liquid: B-3
C-3 or C-4	31/65	64/136	110/214	128/221	305/721		
C-5 or C-6 or C-7	42/65	88/136	129/214	127/221	339/721	Fabric softener	Fabric – liquid: F-1, condensed liquid: F-2, tissue type: F-3
C-8 or C-9	20/65	52/136	69/214	52/221	169/721		
C-10 and C-11	11/65	25/136	42/214	35/221	67/721	Coating agent	Vehicle glass water-repellent use – trigger: CA-1. Aerosol spray: CA-2 Polishing & coating vehicles – trigger: CA-3, wax: CA-4 Polishing & coating vehicle wheels - trigger: CA-5, aerosol spray: CA-6
C-12	7/65	12/136	23/214	23/221	45/721		
B-1 or B-2	16/65	31/136	52/214	65/221	147/721		
B-3	8/65	18/136	25/214	17/221	39/721		
CA-1 or CA-2	21/65	28/136	47/214	47/221	106/721		
AF-1	37/65	73/136	120/214	99/221	359/721	Adhesive	All-purpose – liquid: AD-1, aerosol spray: AD-2
AF-2	15/65	22/136	49/214	54/221	150/721	Air freshener	Freshening indoor & vehicle air - liquid & others: AF-1, aerosol spray: AF-2, diffuser & candle: AF-3
AF-3	11/65	27/136	40/214	33/221	79/721		
D-1 or D-2	50/65	85/136	138/214	148/221	491/721	Deodorant	Fabric – trigger: D-1, aerosol spray: D-2 Air-conditioners & heaters – trigger: D-3, aerosol spray: D-4 Toilet & others - gel/diffuser: D-5
D-3 or D-4	9/65	24/136	36/214	39/221	89/721		
D-5	11/65	18/136	34/214	24/221	75/721		

Note: ages: infant, 0−2 y; toddler, 3−6 y; child, 7−12 y; youth, 13−18 y; adult, 19+ y. Abbreviations: n ^a^, number of families that keep all product ranges; y, years old.

**Table 2 ijerph-16-00733-t002:** Calculated exposed amounts of products.

Product (Mean Volume of Main-Use Product)		g/month (30 days)				g/day			
		Median	50th	75th	95th	Median	50th	75th	95th
C	C-1 (528.9 mL)	38.2	12.3	47.3	537.8	1.3	0.4	1.6	17.9
	C-2 (741 mL)	292.2	115.6	462.4	1632.8	9.7	3.9	15.4	54.4
	C-3 (504.1 mL)	51.5	9.5	39.9	1379.5	1.7	0.3	1.3	46
	C-5 (517.4 mL)	55.7	20.5	94.7	896.4	1.9	0.7	3.2	29.9
	C-6 (651.8 mL)	525.2	231.2	693.6	3460	17.5	7.7	23.1	115.3
	C-8 (486.3 mL)	38.8	10.3	47.3	537.8	1.3	0.3	1.6	17.9
	C-12 (364 mL)	4.5	0.3	3.7	134.5	0.2	0.01	0.1	4.5
	C-1/C-2/C-3/C-5/C-6/C-8/C-12	1006.1	399.7	1388.9	8578.8	33.5	13.3	46.3	286
CA	CA-1 (482.1 mL)	11.3	5.3	24.9	112	0.4	0.2	0.8	3.7
	CA-2 (340.9 mL)	32.2	26	57.3	346.8	1.1	0.9	1.9	11.6
	CA-1/CA-2	43.5	31.3	82.2	458.8	1.5	1	2.7	15.3
AD	AD-1 (22.4 g)	0.1	0.04	0.2	0.6	0.003	0.001	0.007	0.02
	AD-2 (255.6 mL)	2.7	1.3	6.4	31.6	0.09	0.04	0.2	1.1
	AD-1/AD-2	2.8	1.34	6.6	32.2	0.09	0.04	0.2	1.1
AF	AF-1 (177.3 mL)	112.9	40.1	121.4	485.6	3.8	1.3	4	16.2
	AF-2 (291.7 mL)	2.1	0.6	2.1	21.3	0.07	0.02	0.07	0.7
	AF-3 (242 g)	179.1	79.9	242	484	6	2.7	8.1	16.1
	AF-1/AF-2/AF-3	294.1	120.6	365.5	990.9	9.8	4	12.2	33
D	D-1 (443.9 mL)	29.2	4.7	46.3	612.2	1.0	0.2	1.5	20.4
	D-2	25.4	4.6	31.6	677.8	0.8	0.2	1.1	22.6
	D-3	1.0	0.6	1.5	5.1	0.03	0.02	0.05	1.7
	D-4 (318.8 mL)	1.3	0.7	2.5	11.0	0.04	0.02	0.08	0.37
	D-5 (211.7 g)	287.9	175.7	211.7	1481.9	9.6	5.9	7.1	49.4
	D-1/D-2/D-3/D-4/D-5	344.8	186.3	293.6	2788	11.5	6.2	9.8	92.9

**Table 3 ijerph-16-00733-t003:** Toxicity information of studied substances in products.

Substance	CAS RN	Vapor pressure (mmHg, 25 °C) ^c^	RfC/chronic NOAEL (mg/m^3^)	Toxicity (mg/m^3^)	Critical effect/Assessment factor (AF ^g^)
Toluene	108-88-3	28.4	5 (RfC)	NOAEL(ADJ) = 46 ^a^	Neurological effect (10)
Methanol	67-56-1	127	20 (RfC)	POD_Internal_ = 858 mg-h/L ^a^	Reduced brain weight in rat (100)
2-BE	111-76-2	0.88	1.6 (RfC)	BMCL (HEC) = 1.6 ^a^	Hemosiderin deposition in the liver (10)
MIBK	78-93-3	90.6	5 (RfC)	LEC (ADJ) = 1517 ^a^	Developmental toxicity (300)
EB	100-41-4	9.6	1 (RfC)	NOAEL (ADJ) = 434 ^a^	Developmental toxicity (300)
Xylene	1330-20-7	8.3/6.6/8.7	0.1 (RfC)	NOAEL (HEC) = 39 ^a^	Impaired motor coordination (300)
Isopropanol	67-63-0	45.4	7 (RfC)	Inhalation REL = 7 ^b^	Kidney development
Benzene	71-43-2	94.8	0.03 (RfC)	BMCL(ADJ) = 8.2 ^a^	Decreased lymphocyte count (300)
Benzyl alcohol	100-51-6	0.094	31.9 ^*^	4 weeks/rat, NOAEC = 1072 ^d^	AS Factors = 6, sub-acute to chronic: 6 (0.18 ^g^)
EA	141-78-6	93.2	114.3 ^*^	96 d/rat, LOAEL = 1280 ^d^	AS Factors = 6, sub-chronic to chronic: 2, LOAEL to NOAEL: 3 (0.18 ^g^)
1-Butanol	71-36-3	7	136.4 ^*^	13 weeks/rat, NOEL = 1516 ^d^	AS Factors = 2, sub-chronic to chronic: 2 (0.18 ^g^)
PG	57-55-6	0.15	89.3 ^*^	90 d/rat, NOAEC = 1000 ^d^	
2-PE	122-99-6	0.007	1.4 ^*^	2 weeks/rat, NOAEC = 48.2 ^d^	AS Factors = 6, sub-acute to chronic: 6 (0.18 ^g^)
SB	532-32-1	0.0122	0.7 ^*^	4 weeks/rat, NOAEC = 25 ^d^	
MIT	2682-20-4	0.062	0.003 ^*^	90 d/rat, LOAEC = 3.2 ^e^	AS Factors = 2, sub-chronic to chronic: 2 (0.18 ^g^)
OIT	26530-20-1	2.98	0.1 ^*^	90 d/rat, NOAEC = 0.34 ^e^	
DDAC	7173-51-5	<4.3 × 10^−5^	0.0 ^*^	90 d/rat, LOAEC = 0.11 ^f^	
FA	64-18-6	42.59	10.9 ^*^	13 weeks/rat, NOAEC = 244 ^d^	
Hexane	110-54-3	153	881.2 ^*^	16 weeks/rat, LOAEC = 10574.2 ^d^	AS Factors = 2, sub-chronic to chronic: 2 (0.5 ^g^)
Heptane	142-82-5	37.49	3117.5 ^*^	16 weeks/rat, NOAEC = 12470 ^d^	
BA	123-86-4	11.5	214.3	16 weeks/rat, NOAEC = 2400 ^d^	AS Factors =2, sub-chronic to chronic: 2 (0.18 ^g^)

Abbreviations: 2-BE, 2-butoxyethanol; MIBK, methyl isobutyl ketone (4-methyl-2-pentanone); EB, ethylbenzene; 1,2,3-TMB, 1,2,3-trimethylbenzene; EA, ethyl acetate; PG, propylene glycol (1,2-propandiol); 2-PE, 2-phenoxyethanol; SB, sodium benzoate; MIT, 2-methyl-4-isothiazolin-3-one; OIT, 2-n-octyl-4-isothiazolin-3-one; DDAC, didecyldimethylammonium chloride; FA, formic acid; BA, butyl acetate; ADJ, adjusted; UF, uncertainty factor; AF, adjustment factor; BMCL, benchmark concentration 95% lower bound; HEC, human equivalent concentration; REL, recommended exposure level (chronic reference exposure level); NOAEC, no observed adverse effect concentration; LOAEL, lowest observed adverse effect level; NOEL, no observed effect level; NOAEL: no observed adverse effect level; LOAEC, lowest observed adverse effect concentration; AS factor, assessment factor-route to route exploration (ECETOC TR No.86). *^a^* As reported in EPA documents (U.S. EPA IRIS chemical assessment summary) [37]. ^*^ chronic NOAEL. *^b^* As reported in Cal/EPA documents. *^c^* As reported in PubChem (NIH). *^d^* As reported in ECHA dossier (European Chemicals Agency, registration dossier). *^e^* As reported in the EPA RED report. *^f^* As reported in a KOSHA report [38]. *^g^* Adjustment factor: 0.18, 6 h/24 h·5 d/7 d; 0.5, 12 h/24 h·7 d/7 d.

**Table 4 ijerph-16-00733-t004:** Analyzed concentrations of studied substances in products C, B and CA.

Substance (LOQ)	Substance	C (mg/kg)			B	CA (mg/kg)		
		D.P.	Max. (Min.)	Mean	D.P.	D.P.	Max. (Min.)	Mean
Toluene (10)	Toluene	4/109	6.6 × 10^3^ (14)	1.6 × 10^3^	0/13	8/43	1.0 × 10^3^ (36)	3.8 × 10^2^
Methanol (100)	Methanol	3/109	1.8 × 10^4^ (3.2 × 10^3^)	8.3 × 10^3^	0/13	NT	-	-
2-BE (5)	2-BE	23/109	2.5 × 10^5^ (5)	2.4 × 10^4^	0/13	3/43	4.0 × 10^2^ (1.0 × 10^2^)	2.0 × 10^2^
MIBK (50)	MIBK	0/109	<50	-	0/13	0/43	<50	-
EB (10)	EB	NT	-	-	NT	5/43	2.0 × 10^5^ (2.4 × 10^2^)	4.1 × 10^4^
Xylene (10)	Xylene	NT	-	-	NT	5/43	9.1 × 10^4^ (1.8 × 10^3^)	2.2 × 10^4^
Isopropanol (50)	Isopropanol	9/109	4.8 × 10^5^ (1.0 × 10^2^)	9.0 × 10^4^	0/13	9/43	8.3 × 10^5^ (30)	1.4 × 10^5^
Benzene (1)	Benzene	19/1421	1.5 × 10^4^ (3.5)	8.4 × 10^2^	0/100	0/43	<10	-
Benzyl alcohol (100)	Benzyl alcohol	NT	-	-	NT	NT	-	-
EA (50)	EA	NT	-	-	NT	NT	-	-
1-Butanol (50)	1-Butanol	5/109	6.9 × 10^2^ (58)	2.6 × 10^2^	0/13	NT	-	-
PG (50)	PG	24/109	3.1 × 10^5^ (10)	3.4 × 10^4^	NT	NT	-	-
2-PE (5)	2-PE	5/109	2.3 × 10^3^ (14)	9.1 × 10^2^	NT	NT	-	-
SB (5)	SB	7/109	6.3 × 10^3^ (8)	3.4 × 10^3^	NT	7/43	6.7 × 10^3^ (2.0 × 10)	2.2 × 10^3^
MIT (1)	MIT	4/109	8.4 × 10 (21)	6.3 × 10	NT	NT	-	-
OIT (1)	OIT	1/109	2.1 × 10	2.1 × 10	NT	0/43	<1	-
DDAC (1)	DDAC	2/109	6.3 × 10^2^ (1.5 × 10^2^)	3.9 × 10^2^	NT	NT	-	-
FA (5)	FA	20/109	9.1 × 10^3^ (11)	5.2 × 10^2^	NT	NT	-	-
Hexane (5)	Hexane	NT	-	-	NT	NT	-	-
Heptane (10)	Heptane	NT	-	-	NT	2/43	2.0 × 10^2^ (2.0 × 10)	2.0 × 10^2^
BA (10)	BA	NT	-	-	NT	1/43	1.0 × 10^4^	1.0 × 10^4^

Abbreviations: LOQ, limit of quantitation; C, cleaner; B, bleaching agent; NT, not tested; D.P., number of products with detects/total tested products; CA, coating agent.

**Table 5 ijerph-16-00733-t005:** Analyzed concentrations of studied substances in products AD, AF and D.

Substance (LOQ)	AD (mg/kg)			AF (mg/kg)			D (mg/kg)		
	D.P.	Max. (Min.)	Mean	D.P.	Max. (Min.)	Mean	D.P.	Max. (Min.)	Mean
Toluene (10)	NT	-	-	NT	-	-	NT	-	-
Methanol (100)	1/30	8.6 × 10^5^	8.6 × 10^5^	0/54	<100	-	1/208	1.5 × 10^3^	1.5 × 10^3^
2-BE (5)	NT	-	-	2/54	4.9 × 10^3^ (5.9 × 10^2^)	2.7 × 10^3^	1/47	286	2.8 × 10^2^
MIBK (50)	3/30	4.5 × 10^5^ (1.0 × 10^2^)	1.5 × 10^5^	NT	-	-	NT	-	-
EB (10)	1/30	5.9 × 10	5.9 × 10	NT	-	-	NT	-	-
Xylene (10)	0/30	<100	-	NT	-	-	NT	-	-
Isopropanol (50)	5/30	2.9 × 10^3^ (64)	8.4 × 10^3^	1/54	4.9 × 10^3^	4.9 × 10^3^	2/47	2.3 × 10^5^ (27)	1.1 × 10^5^
Benzene (1)	NT	-	-	3/4593	1.3 (1.2)	1.27	7/482	1.3	0.1
Benzyl alcohol (100)	1/30	5.6 × 10^2^	5.6 × 10^2^	NT	-	-	NT	-	-
EA (50)	2/30	1.3 × 10^3^ (1.0 × 10^2^)	7.0 × 10^2^	NT	-	-	NT	-	-
1-Butanol (50)	NT	-	-	NT	-	-	NT	-	-
PG (50)	NT	-	-	NT	-	-	NT	-	-
2-PE (5)	1/30	1.4 × 10^4^	1.4 × 10^4^	0/54	<5	-	4/47	1.0 × 10^4^ (5.8 × 10^3^)	7.0 × 10^3^
SB (5)	NT	-	-	2/54	1.9 × 10^3^ (6.9 × 10^2^)	1.3 × 10^3^	6/47	9.4 × 10^3^ (52)	2.5 × 10^3^
MIT (1)	NT	-	-	0/54	<1	-	1/47	1.9	1.9
OIT (1)	0/30	<1	-	0/54	<1	-	0/47	<1	
DDAC (1)	NT	-	-	0/54	<1	-	3/47	1.2 × 10^3^ (2.8)	1.9 × 10^2^
FA (5)	NT	-	-	0/54	<5	-	1/47	6.5 × 10^2^	6.5 × 10^2^
Hexane (5)	3/30	3.0 × 10^4^ (1.9 × 10^4^)	2.5 × 10^4^	NT	-	-	NT	-	-
Heptane (10)	NT	-	-	NT	-	-	NT	-	-
BA (10)	1/30	1.0 × 10^2^	1.0 × 10^2^	NT	-	-	NT	-	-

Abbreviations: LOQ, limit of quantitation; AD, adhesive; AF, air-freshener; D, deodorant.

**Table 6 ijerph-16-00733-t006:** Worst-case combined exposure concentrations via inhalation.

Substance	Infants (worst-case exposure per month)				Toddlers		Children		Youths	
	C_a_ (eq 1)	W_f_	C_inh con_ (eq 2)	C_inh amount_ (eq 3)	C_inh con_	C_inh amount_	C_inh con_	C_inh amount_	C_inh con_	C_inh amount_
Toluene	3.5 × 10^−2^	0.2 × 10^−2^	1.2 × 10^−2^	7.1 × 10^−2^	0.1 × 10^−1^	1.7 × 10^−1^	7.6 × 10^−3^	0.4 × 10^−2^	7.6 × 10^−3^	0.2 × 10^−2^
Methanol	1.5 × 10^−1^	8.8 × 10^−1^	5.1	31	4.2	7.3	3.3	1.6	3.3	8.7 × 10^−1^
2−BE	4.8 × 10^−1^	2.8 × 10^−2^	1.6 × 10^−1^	9.8 × 10^−1^	1.3 × 10^−1^	2.3 × 10^−1^	0.1	5.1 × 10^−2^	0.1	2.8 × 10^−2^
MIBK	2.6	1.5 × 10^−1^	8.7 × 10^−1^	5.2	7.1 × 10^−1^	1.2	5.6 × 10^−1^	2.7 × 10^−1^	5.6 × 10^−1^	1.4 × 10^−1^
EB	7.3 × 10^−1^	4.2 × 10^−2^	2.4 × 10^−1^	1.4	0.2	3.5 × 10^−1^	1.5 × 10^−1^	7.7 × 10^−2^	1.5 × 10^−1^	4.1 × 10^−2^
Xylene	3.8 × 10^−1^	2.2 × 10^−2^	1.2 × 10^−1^	7.7 × 10^−1^	0.1	1.8 × 10^−1^	8.3 × 10^−2^	0.4 × 10^−1^	8.3 × 10^−2^	2.2 × 10^−2^
Isopropanol	6.3	3.6 × 10^−1^	2.1	12	1.7	3.0	1.3	6.6 × 10^−1^	1.3	3.5 × 10^−1^
Benzene	1.4 × 10^−1^	0.8 × 10^−2^	4.7 × 10^−2^	2.8 × 10^−1^	6.7 × 10^−1^	6.7 × 10^−2^	0.3 × 10^−1^	1.5 × 10^−2^	0.3 × 10^−1^	0.8 × 10^−1^
Benzyl alcohol	0.9 × 10^−2^	0.5 × 10^−3^	0.3 × 10^−2^	1.8 × 10^−2^	0.2 × 10^−2^	0.4 × 10^−2^	0.2 × 10^−2^	0.9 × 10^−3^	0.2 × 10^−2^	0.5 × 10^−3^
EA	1.2 × 10^−2^	0.7 × 10^−3^	0.4 × 10^−2^	2.4 × 10^−2^	0.3 × 10^−2^	0.6 × 10^−2^	0.3 × 10^−2^	0.1 × 10^−2^	0.3 × 10^−2^	0.7 × 10^−3^
1-Butanol	0.3 × 10^−2^	0.2 × 10^−3^	0.001.2 × 10^−3^	0.6 × 10^−2^	0.1 × 10^−2^	0.1 × 10^−2^	0.1 × 10^−2^	0.3 × 10^−3^	0.1 × 10^−2^	0.2 × 10^−3^
PG	5.9 × 10^−1^	3.4 × 10^−2^	1.9 × 10^−1^	1.1	1.6 × 10^−1^	2.8 × 10^−1^	1.2 × 10^−1^	6.2 × 10^−2^	1.2 × 10^−1^	3.3 × 10^−2^
2-PE	3.8 × 10^−1^	0.2 × 10^−1^	1.2 × 10^−1^	7.7 × 10^−1^	1.0 × 10^−1^	1.8 × 10^−1^	0.8 × 10^−1^	0.4 × 10^−1^	0.8 × 10^−1^	0.2 × 10^−1^
SB	1.5 × 10^−1^	0.96 × 10^−2^	5.2 × 10^−2^	3.1 × 10^−1^	4.3 × 10^−2^	7.5 × 10^−2^	3.4 × 10^−2^	1.6 × 10^−2^	3.4 × 10^−2^	0.9 × 10^−2^
MIT	0.1 × 10^−2^	0.6 × 10^−4^	0.3 × 10^−3^	0.2 × 10^−2^	0.3 × 10^−3^	0.5 × 10^−3^	0.2 × 10^−3^	0.1 × 10^−3^	0.2 × 10^−3^	0.6 × 10^−4^
OIT	0.3 × 10^−3^	0.2 × 10^−4^	0.1 × 10^−3^	0.6 × 10^−3^	0.1 × 10^−3^	0.1 × 10^−3^	0.1 × 10^−3^	0.3 × 10^−4^	0.1 × 10^−3^	0.2 × 10^−4^
DDAC	0.9 × 10^−2^	0.5 × 10^−3^	0.3 × 10^−2^	1.8 × 10^−2^	0.2 × 10^−2^	0.4 × 10^−2^	0.2 × 10^−2^	0.9 × 10^−3^	0.2 × 10^−2^	0.5 × 10^−3^
FA	1.9 × 10^−2^	1.1 × 10^−3^	0.6 × 10^−2^	3.8 × 10^−2^	0.5 × 10^−2^	0.9 × 10^−2^	0.4 × 10^−2^	0.2 × 10^−2^	0.4 × 10^−2^	0.1 × 10^−2^
Hexane	4.4 × 10^−1^	2.5 × 10^−2^	1.4 × 10^−1^	8.9 × 10^−1^	1.2 × 10^−1^	2.1 × 10^−1^	9.5 × 10^−2^	4.7 × 10^−2^	9.5 × 10^−2^	2.5 × 10^−2^
Heptane	4.4 × 10^−1^	2.5 × 10^−2^	1.4 × 10^−1^	9.0 × 10^−1^	1.2 × 10^−1^	2.1 × 10^−1^	9.7 × 10^−1^	4.7 × 10^−1^	9.7 × 10^−1^	2.5 × 10^−2^
BA	1.7 × 10^−1^	1.0 × 10^−2^	5.9 × 10^−2^	3.5 × 10^−1^	4.8 × 10^−2^	8.4 × 10^−2^	3.8 × 10^−2^	1.8 × 10^−2^	3.8 × 10^−2^	0.1 × 10^−1^
Values	Ap (g/use/month): C-1/2/3/5/6/8/12: 1006.1, CA-1/2: 43.5, AD-1/2: 2.8, AF-1/2/3: 294.1, D-1/2/3/4/5 344.8, sum: 1691.3 V (m3): 33.30 (living room), 30.3 (bedroom), 9.3 (toilet), 24.5 (kitchen) sum: 97.4 N (-h): 0.5 (living room), 1 (bed room), 2 (toilet), 2.5 (kitchen), General ventilation ratio (home total): 0.6, sum: 0.6·24 h·30 d = 432 (month) t (h/month): mean staying time of infant in home, 16 h/day sum: 16 h·30 d = 480 Abs: 1 (100%) AF: 1 IR-day (m^3^/day-kg, mean): 0.87 (M)/ 0.90(F), mean: 0.89, used value: 0.89/24 h = 0.037 BW (mean kg): 5.9 (0–3 month), 8.0 (3–6 month), 8.9 (6–9 month), 10.1 (9–12 month), 11.3 (12–24 month), mean: 8.8 W_f_: proportion of mean con. of substances in products						t-age (h): infant: 480 (16 h·30 d) toddler: 396 (12 h·21 d+16 h·9 d) child: 312 (8 h·21 d+16 h·9 d) youth: 312 (8 h·21 d+16 h·9 d) IR-age: infant: 0.89/24 h = 0.037 (m^3^/ h-kg) toddler: 0.55/24 h = 0.023 (m^3^/ h-kg) child: 0.295/24 h = 0.0123 (m^3^/ h-kg) youth: 0.255/24 h = 0.0106 (m^3^/ h-kg) BW-age (kg): infant mean: 8.8 toddler mean: 19.05 ((19.6(M)+18.5(F))/2) child mean: 36.55 (30.15 (7–9 y), 42.95 (10–12 y)) youth: 58.65 (55.95 (13–15 y), 61.35 (16–18 y))			

Abbreviations: C_a_, concentration of the substance in the air; A_p_, amount of product used; W_f_, ratio of mean concentration of substance in product; V, volume of space; t, duration of use (indoor time in the home); AF, air-born fraction; C_inh con_, exposure concentration by inhalation; abs, absorption ratio to body; n, frequency of use; IR, resting inhalation rate; BW, body weight; C_inh amount_, exposure amount by inhalation.

**Table 7 ijerph-16-00733-t007:** Indoor times and products used by the surveyed families.

Subjects	Survey (*n* = 1175 Family, % Ratio)	Indoor Time at Home: No. of Subject		Inhalation Rate (m^3^/day-kg)		No. of Families Used Household Products			
		Working Day	Non-Working Day	Mean	95th	C (/n, Subject)	S/B (/n, Subject)	F/CA (/n, Subject)	AD/AF/D (/n, Subject)
Infant (0−2) (*n* = 72 infants)	65 families (4.7%) with 72 infants (M:43/F:29)	~8 h: 12/72 8~16 h: 20 16~24 h: 40	~8 h: 3/72 8~16 h: 16 16~24 h:53	0.87 (M)	1.32	C-1:37/72, C-2:18, C-3:29, C-4:19, C-5:11, C-6:17, C-7:33, C-8:13, C-9:13, C-10:24, C-11:21, C-12:7	S-1:39/72, S-2:55, S-3:34, S-4:14, S-5:40, S-6:9 B-1:10, B-2:12, B-3:8	F-1:44/72, F-2:25, F-3:8 CA-1:17, CA-2:13, CA-3:12, CA-4:11, CA-5:8, CA-6:7	AD-1:39/72, AD-2:5, AF-1:37, AF-2:15, AF-3:11 D-1:49, D-2:18, D-3:7, D-4:6, D-5:11
				0.90 (F)	1.78				

Toddler (3−6) (n = 158 toddlers)	136 families (11.6%) with 158 toddlers (M:70/F:88)	~8 h: 27/158 8~16 h: 78 16~24 h: 53	~8 h: 18/158 8~16 h: 37 16~24 h:103	0.57 (M)	0.73	C-1:59/158, C-2:43, C-3:54, C-4:41, C-5:21, C-6:26, C-7:76, C-8:27, C-9:37, C-10:51, C-11:44, C-12:12	S-1:69/158, S-2:111, S-3:70, S-4:20, S-5:78, S-6:18 B-1:15, B-2:27, B-3:18	F-1:95/158, F-2:53, F-3:11 CA-1:21, CA-2:17, CA-3:20, CA-4:15, CA-5:13, CA-6:10	AD-1:80/158, AD-2:13, AF-1:73, AF-2:22, AF-3:27 D-1:79, D-2:31, D-3:13, D-4:17, D-5:18
				0.53 (F)	0.70				

Young child (7−9) (n = 114 young children)	109 families (9.3%) with 114 young children (M:64/F:50)	~8 h: 22/114 8~16 h: 69 16~24 h: 23	~8 h: 13/114 8~16 h: 37 16~24 h: 64	0.40 (M)	0.55	C-1:52/114, C-2:38, C-3:50, C-4:28, C-5:18, C-6:24, C-7:55, C-8:24, C-9:26, C-10:41, C-11:36, C-12:11	S-1:64/114, S-2:81, S-3:57, S-4:18, S-5:62, S-6:10 B-1:17, B-2:21, B-3:24	F-1:79/114, F-2:39, F-3:12 CA-1:19, CA-2:15, CA-3:25, CA-4:15, CA-5:15, CA-6:15	AD-1:64/114, AD-2:12, AF-1:65, AF-2:26, AF-3:23 D-1:66, D-2:23, D-3:11, D-4:14, D-5:15
				0.36 (F)	0.47				

Child (10−12) (n = 116 children)	105 families (8.9%) with 116 children (M:56/F:60)	~8 h: 24/116 8~16 h: 76 16~24 h: 16	~8 h: 8/116 8~16 h: 38 16~24 h: 70	0.30 (M)	0.35	C-1:54/116, C-2:42, C-3:44, C-4:34, C-5:15, C-6:29, C-7:49, C-8:16, C-9:22, C-10:45, C-11:45, C-12:12	S-1:59/116, S-2:75, S-3:46, S-4:10, S-5:61, S-6:14 B-1:11, B-2:15, B-3:18	F-1:74/116, F-2:37, F-3:4 CA-1:22, CA-2:10, CA-3:20, CA-4:14, CA-5:15, CA-6:8	AD-1:62/116, AD-2:7 AF-1:55, AF-2:23, AF-3:17 D-1:66, D-2:29, D-3:6, D-4:15, D-5:19
				0.29 (F)	0.32				

Young youths (13−15) (n = 105 young youth)	98 families (8.3%) with 105 young youth (M:50/F:55)	~8 h: 23/105 8~16 h: 75 16~24 h: 7	~8 h: 11/105 8~16 h: 29 16~24 h: 65	0.27 (M)	0.29	C-1:40/105, C-2:36, C-3:52, C-4:35, C-5:17, C-6:23, C-7:42, C-8:17, C-9:11, C-10:43, C-11:31, C-12:12	S-1:69/105, S-2:59, S-3:33, S-4:19, S-5:56, S-6:9 B-1:14, B-2:21, B-3:16	F-1:68/105, F-2:29, F-3:8 CA-1:18, CA-2:14, CA-3:14, CA-4:14, CA-5:12, CA-6:7	AD-1:58/105, AD-2:4, AF-1:51, AF-2:26, AF-3:19 D-1:62, D-2:26, D-3:6, D-4:17, D-5:13
				0.24 (F)	0.27				

Youths (16−18) (*n* = 134 youth)	123 families (10.5%) with 134 youth (M:84/F:50)	~8 h: 34/134 8~16 h: 98 16~24 h: 2	~8 h: 17/134 8~16 h: 56 16~24 h: 61	0.24 (M)	0.28	C-1:53/134, C-2:56, C-3:49, C-4:37, C-5:16, C-6:21, C-7:45, C-8:19, C-9:12, C-10:45, C-11:33, C-12:11	S-1:91/134, S-2:73, S-3:38, S-4:24, S-5:53, S-6:12 B-1:15, B-2:27, B-3:13	F-1:77/134, F-2:32, F-3:10 CA-1:18, CA-2:13, CA-3:13, CA-4:15, CA-5:8, CA-6:5	AD-1:59/134, AD-2:8, AF-1:48, AF-2:28, AF-3:14 D-1:74, D-2:32, D-3:7, D-4:16, D-5:11
				0.24 (F)	0.30				

Adults (19−) (*n* = 2032 adults)	721 families (61.4%) with 2032 adults (M:952/F:1080)	Employed-male ~8 h: 205/651 8~16 h: 430 16~24 h: 16 Employed-female ~8 h: 131/551 8~16 h: 395 16~24 h: 25	Employed-male ~8 h: 114/651 8~16 h: 274 16~24 h: 263 Employed-female ~8 h: 70/551 8~16 h: 217 16~24 h: 264	0.23 (M)	0.26	C-1:276/2032, C-2:216, C-3:252, C-4:148, C-5:74, C-6:121, C-7:266, C-8:114, C-9:85 C-10:177, C-11:141, C-12:45	S-1:459/2032, S-2:436, S-3:239, S-4:101, S-5:370, S-6:48 B-1:61, B-2:108, B-3:39	F-1:463/2032, F-2:25, F-3:8 CA-1:80, CA-2:53, CA-3:53, CA-4:63, CA-5:39, CA-6:34	AD-1:352/2032, AD-2:40 AF-1:359, AF-2:150, AF-3:79 D-1:459, D-2:166, D-3:29, D-4:76, D-5:75
				0.23 (F)	0.25				

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
