# Peer review of "Combined Assessment of Preschool Childrens’ Exposure to Substances in Household Products"

_ijerph, 2019, doi:10.3390/ijerph16050733_

Round 1
Reviewer 1 Report
Thus study showed assessing the combined exposure to infants and toddlers by inhalation for 1175 families by using determinng the mean concentrations household products in order to identify respiratory health risks. Risks to infants and toddlers should be assessed on the basis of the toxicity values of combined exposed hazardous substances, as well as their combined concentrations and amounts. This exposure assessment approach might be used to establish improved guidelines for products that may pose inhalation hazards to infants and toddlers. Their focus on combined products is very important valuable studies. However, it is unclear what is anew finding in this analysis. Authors should state how large amount of risk is estimated exposure amounts for combined products according to this analysis in Abstract, Results or Discussion.
Author Response
Response to Reviewer 1 Comments
Point 1: This exposure assessment approach might be used to establish improved guidelines for products that may pose inhalation hazardous to infants and toddlers. Their focus on combined products is very analysis. However, it is unclear what is a new finding in this exposure amounts for combined products according to this analysis in Abstract, Results or Discussion.
Response 1: Thank you very much for your comments. Your detailed and very valuable comments improved the quality and presentation of this manuscript. I appreciate your time and effort. I answered to your questions and comments.
This study is about ‘exposure assessment study of inhalable consumer products and their ingredients by household use via respiratory route to adult users, by-standing second exposure such as preschool children and children’. As a further study, we are investigating health risk assessment study to preschool children and children from household product use by primary users. The lack of products exposure information and exposure assessment was a major limitation of the risk assessment study.
This study was carried out from step 1 to step 4.
1 step: identification of hazardous substances (household products): Sample preparation of products.
2 step: analysis of hazardous substances in target products: Identification and analysis of ingredients in products.
3 step: selecting toxicological information of target substances & hazardous identification
4 step: exposure assessment (determining exposure factors to studied subjective)
Next step is risk assessment study, we are carrying out risk assessment study to preschool children and children.
We followed NIER notification (NIER, No. 2017-55) about the procedure of exposure and risk assessment.

Reviewer 2 Report
This is an interesting manuscript that aims to assess combined exposure levels to substances in common household products among children in Korea. The data and exposure assessment paradigm used have potential value for future risk assessments of these exposures. While the paper is generally well-written, I have several comments relating to the Methods and interpretation of the data:
It is stated that households for the survey were selected to be representative of actual product users in Korea, but few details are provided about the sampling methods. What was the participation rate and is it possible that people included in the study may be more likely to use higher amounts of certain products (reflecting the "worst-case scenario" as described in the paper)?
The survey is described as collecting information on type of house, average house size, among other characteristics in addition to "qualitative descriptions" of product use habits. It is unclear to me to what extent these variables contribute to the described exposure assessment models (for example, is it taken into account whether the child is actually in the home when a certain product is usually applied?). How are these qualitative use habits integrated in the exposure assessment?
Is it possible to conduct validation of the model-derived indoor air levels of certain pollutants with actual indoor monitoring for a small subset of the included homes? This would provide valuable insight to the performance of this model-based approach for future exposure assessments.
Certain parts of the Discussion talking about resulting health effects (e.g. pg. 21 line 81) and respiratory effects (pg. 22 line 120) may be somewhat overstated since the current study does not actually conduct a health risk assessment (as acknowledged on pg. 21 line 77). Some discussion about how the modeled exposures relate to published data on health endpoints (including from animal models) would be informative. Also, would suggest some discussion (and potential limitations/caveats) about how this approach could be applied to future epidemiological studies.
Minor comments:
In section 2.4, please provide the rationale for treating concentrations of substances <LOQ as zero.
The title, abstract, and tables are somewhat disjointed with respect to describing the study population (i.e. title says "preschool children" and abstract says "..combined exposure to infants and toddlers", but paper also describes older age-groups, e.g. 16-18 years old).
Author Response
Response to Reviewer 2 Comments
Point 1: It is stated that households for the survey were selected to representative of actual product users in Korea, but few details are provides about the sampling methods. What was the participation rate and is it possible that people included in the study may be more likely to use higher amounts of certain products (reflecting the “worst-case scenario” as described in the paper)?
Response 1: Thank you very much for your comments. Your detailed and very valuable comments improved the quality and presentation of this manuscript. I appreciate your time and effort. I answered to your questions and comments.
About survey methods, I revised and inserted sentences „2. Materials and Methods, 2.1 Survey‟ part as followed;
2.1 Survey
A market searching was conducted to elucidate which studied household products were commonly used in the Korean market. Survey-company carried out the market searching of the currently used household products. The web searching collected current information on the products. In our previous study, in order to obtain data on the exposure of consumer to household products, web surveys were carried out [41]. A total of 1175 households participated in this web-survey study completed the questionnaire of survey. The purpose of web-survey was to obtain information about the household exposure of the studied products at home. Survey was carried out a Korean survey company. Survey-company had participants fool in all provinces and cities in Korea. At first, an e-mail was sent to 12 times ~ 15 times the participants of the targeted 1000 cases. If participants agreed to take the survey among them answered that have experience using at least one product in studied products, a web link was sent to them. The web-survey questionnaires consisted of purchasing/using information of products as follows: list of products used at home and the frequency of use, estimation of quantitative exposed duration to products, quantitative amount of products used, and demographic data. Furthermore, the survey questionnaire included the questions on family members of respondent and on combined multiple-use of products. Combined exposed amounts of the studied families by household products at home were evaluated.
In this study, „worst-case scenario‟ means the considering worst case exposure time and amounts‟ as follow;
Considering the worst-case exposure, exposure time and exposure amounts to preschool children and children by household products use was assumed to duration time at home and to exposed amounts during staying at home.
Primary exposure to products occurs to the user who actively uses the household products. Secondary exposure is exposure that may occur after the actual use or application of the product. The user of products may be subject to both primary and secondary exposure whereas the non-user or bystander will only experience secondary exposure. Primary exposures are invariably higher than secondary exposures, however, some specific subgroups of the population may experience higher secondary exposures because of their specific behavior (e.g, children crawling on a treated carpet) (ECHA, 2013).
Point 2: The survey is described as collecting information on type of house, average house size, among other characteristics in addition to qualitative description of product use habits. It
is unclear to me to what extent these variables contribute to the described exposure assessment models (for example, is it taken into account whether the child is actually in the home when a certain product is usually applied?). How are these qualitative use habits integrated in the exposure assessment?
Response 2: Thank you very much for your comments. Your detailed and very valuable comments improved the quality and presentation of this manuscript. I appreciate your time and effort. I answered to your questions and comments.
Even though survey results included information of house size, each room size, ventilation rate, etc, exposure factors for exposure assessment were sourced from the Korean Exposure Factors Handbook (Jang et al, 2008). And survey results included the staying time and exposed amounts of preschool children and children. Considering the worst-case exposure, exposure time and exposure amounts to preschool children and children by household products use was assumed to duration time at home and to exposed amounts during staying at home.
Point 3: Is it possible to conduct validation of the model-derived indoor air of certain pollutants with actual indoor monitoring for small subset of the included homes? This would provide valuable insight to the performance of this model-based approach for future exposure assessments.
Response 3: Thank you very much for your comments. Your detailed and very valuable comments improved the quality and presentation of this manuscript. I appreciate your time and effort. I answered to your questions and comments.
In this study, the indoor concentrations of the substances were assumed to be calculated from equation 1 using the mean concentration of the substances in the various products. However, the actual concentrations of the substances in indoor should be analyzed according to the emitted amounts of these substances from the various products [2].
As a further study, we are investigating the actual amount of several chemicals from used consumer products in indoor air. And in order to more accurate exposure assessment study, we are considering several exposure-model based risk assessment study.
Point 4: Certain parts of the Discussion talking about resulting health effects and respiratory effects may be somewhat overstated since the current study does not actually conduct a health risk assessment. Some discussion about how the modelled exposures relate to published data on health endpoints would be informative. Also, would suggest some discussion about how this approach could be applied to future epidemiological studies.
Response 4: Thank you very much for your comments. Your detailed and very valuable comments improved the quality and presentation of this manuscript. I appreciate your time and effort. I answered to your questions and comments.
As your comments, I revised this sentence.
This study considered respiratory exposure by using inhalable consumer products at home. Therefore, we summarized reference toxicity values, inhalation related, RfC and chronic NOAEL. As a further study, we are considering dermal exposure assessment.
Point 5: In section 2.4, please provide the rationale for treating concentrations of substances <LOQ as zero.
Response 5: Thank you very much for your comments. Your detailed and very valuable comments improved the quality and presentation of this manuscript. I appreciate your time and effort. I answered to your questions and comments.
As your comments, I revised this sentence. LOQ was defined as the lowest concentration at which the analyte can not only be reliably detected at which some predefined goals for bias and imprecision are met; concentrations below the LOQ were assigned a value of zero during data analyses.
Point 6: The title, abstract, and tables are somewhat disjointed with respect to describing the study population.
Response 6: Thank you very much for your comments. Your detailed and very valuable comments improved the quality and presentation of this manuscript. I appreciate your time and effort. I answered to your questions and comments.
As your comments, I revised these sentences. In abstract, I revised „preschool children and children infants and toddlers‟.

Reviewer 3 Report
Thank you for inviting me to review the paper "combined Exposure Assessment of preschool Children Substance in Household Products". In this paper, the authors aim to study the concentrations of substances in household products and assessed the combined use of these products, as well as combined inhalation exposure to infants and toddlers based on “the worst-case scenario”
In overall, the study used exposure assessment model to characterize real-life situations involving infants, toddlers, children, and youths with the household products. The result may be helpful to identify the potentially exposed populations, potential exposure pathways, the magnitude, frequency, duration, and temporal contact patterns. This comprehensive study provides motivation for the mandatory regulation of household products.in the future
However, there are some concerns that the authors should consider to address to improve the quality of the paper. My comments are below:
1. There are many previous studies about exposure of the household products. In this study, the authors did not summary previous study results and did not give information about differences between the previous study and this research work.
2. The author used the EAP deterministic exposure assessment method in this study, it would be better if authors combine the probabilistic risk assessment (PRA) technique method in this study.
3. The author should logically provide the aim of this research and explain its importance aspects.
Author Response
Response to Reviewer 3 Comments
Point 1: There are many previous studies about exposure of the household products. In this study, the authors did not summary previous study results and did not give information about differences between the previous study and this research work.
Response 1: Thank you very much for your comments. Your detailed and very valuable comments improved the quality and presentation of this manuscript. I appreciate your time and effort. I answered to your questions and comments.
Survey results of this study classified nearly all application types of studied product categories in Korean market. Many studies about single-use exposure factor of consumer products, cosmetics, household products. Actually, we could not find the similar exposure factors survey studies about diversely classified application types of same products to compare directly. Furthermore, I could not find the similar combined exposure studies about diversely classified application types of same products to compare directly.
Point 2: The author used the EPA deterministic exposure assessment method in this study, it would be better if authors combine the probabilistic risk assessment (PRA) technic method in this study.
Response 2: Thank you very much for your comments. Your detailed and very valuable comments improved the quality and presentation of this manuscript. I appreciate your time and effort. I answered to your questions and comments.
This study followed “Korean National Law Information Center (KNLIC), 2017. Regulation of concerning the way of risk assessment for risk-concerned products (NIER, No. 2017-55)” as exposure assessment method.
Reliable exposure factors from single use including frequency of use, amount of use per application, and duration of use were established in notification by the law by the National Institute of Environment Research (NIER). Furthermore, NIER developed exposure equations and enacted guidelines of exposure- & risk-assessment procedures from single use household product to human health as notification by the law. Still now, user information is limited to adult users, non-occupational consumers, and occupational users. There are not enough exposure information of preschool children and children. These areas are needed to further studies.
According to this notification, the Korean Ministry of Environment (MOE) and NIER conducted exposure assessment and human risk assessment studies to assess hazardous ingredients, most of which were used in consumer products.
As your comments, we will consider the probabilistic risk assessment (PRA) technic method as a further study.
Point 3: The author should logically provide the aim of this research and explain its importance aspects.
Response 3: Thank you very much for your comments. Your detailed and very valuable comments improved the quality and presentation of this manuscript. I appreciate your time and effort. As your comments, I revised conclusion part.
This study investigated a fundamental approach to assess human exposure to household products used in daily life. The process of assessing exposure to household products used in home requires determining the patterns of use, exposure routes, and quantifying potential ingredients intake. This study determined the recent exposure factors of household products used by respondents group with preschool children, respondents group with children, respondents group with youths, and respondents group with only adults. To protect preschool children and children from several hazardous substances ~
